# Therapeutic Landscapes and Psychiatric Care Facilities: A Qualitative Meta-Analysis

**DOI:** 10.3390/ijerph19031490

**Published:** 2022-01-28

**Authors:** Lydia Oeljeklaus, Hannah-Lea Schmid, Zachary Kornfeld, Claudia Hornberg, Christine Norra, Stefan Zerbe, Timothy McCall

**Affiliations:** 1Medical School OWL, Bielefeld University, 33615 Bielefeld, Germany; lydia.oeljeklaus@uni-bielefeld.de (L.O.); hannah-lea.schmid@uni-bielefeld.de (H.-L.S.); claudia.hornberg@uni-bielefeld.de (C.H.); 2Department of General Internal Medicine and Psychosomatics, Heidelberg University Hospital, 69120 Heidelberg, Germany; 3LWL-Hospital Paderborn, Psychiatry Psychotherapy Psychosomatic, 33098 Paderborn, Germany; zachary.kornfeld@lwl.org (Z.K.); christine.norra@lwl.org (C.N.); 4Medical Faculty, Ruhr University Bochum, 44801 Bochum, Germany; 5Faculty of Science and Technology, Free University of Bozen-Bolzano, 39100 Bolzano, Italy; stefan.zerbe@unibz.it

**Keywords:** physical, built, natural, social and symbolic environment, mental disorders, mental health, psychiatric hospital, review, meta-synthesis

## Abstract

The environment in healthcare facilities can influence health and recovery of service users and furthermore contribute to healthy workplaces for staff. The concept of therapeutic landscapes seems to be a promising approach in this context. The aim of this qualitative meta-analysis is to review the effects of therapeutic landscapes for different stakeholders in psychiatric care facilities. A systematic literature search was conducted in the four data bases PubMed, PsycInfo, CINAHL, and Web of Science. Thirteen predominately qualitative studies were included in this qualitative meta-analysis. The methodological quality of these qualitative studies was assessed, using an adapted version of the Journal Article Reporting Standards for Qualitative Research, and a thematic analysis was conducted. The results were categorised into the three main themes of the physical (built and natural), social, and symbolic dimensions of the therapeutic landscape. Given the heterogeneity of the summarised data and an overall methodological quality of the included studies that can be rated as medium, the results should be interpreted with caution. Current findings are based almost exclusively on qualitative studies. Therefore, there is a need for quantitative study designs that investigate the relationship between specific environmental elements and mental health outcomes for different stakeholders in psychiatric facilities.

## 1. Introduction

### 1.1. Environment, Health, and Well-Being

The environment has a significant impact on health [1,2]. Global research emphasises the positive relationship between direct experiences of natural environments and a wide range of health benefits [1,3,4,5], in addition to the key role nature plays in creating healthy environments [6,7], and in relation to an overall healthy society [8]. Direct and continuous contact with nature can be preventive and health promoting [3,4,9], e.g., by increasing physical and mental well-being or reducing stress [10,11,12,13,14,15,16,17], and promoting social inclusion and social cohesion [5,18]. In addition, ecosystems provide elemental services to humans, such as providing resources, regulating clean air and water, supporting nutrient cycling, and providing opportunities for cultural and recreational experiences [19]. Ecosystems can therefore have an impact on mental health (MH) [20], e.g., by providing spaces for physical activity and social contact [21,22].

### 1.2. The Concept of Therapeutic Landscapes

In the context of a healthy environment, the concept of therapeutic landscapes (TL) was introduced and coined by Gesler from 1992 [9]. The aim is to explore why certain environments appear to contribute to a “healing sense of place” [9,23]. Healing environments have been defined as TL, “where the physical and built environments, social conditions and human perceptions combine to produce an atmosphere which is conducive to healing” [24] (p. 96). Within the concept, four overarching elements can be summarised that are characteristic of a TL [3], namely: (1) natural environments (e.g., green and blue spaces), (2) artificial/built environments (e.g., design features), (3) social environments (including sense of place, attitudes, and values) and (4) symbolic environments (including regional identity, religious places). Therefore, it is important to understand the physical and social health-promoting qualities of a given space, and also the more subjective ways in which people might interpret and use it differently [25,26,27].

TL have been examined at different environmental levels, from large-scale (e.g., countryside), to mesoscale (e.g., urban parks), and microscale environments (e.g., hospitals and clinics, gardens, buildings). Additionally, the scope has been further refined by focusing on diverse populations (e.g., different age groups, gender, cultures, physical abilities, and place-specific practices) [25,27]. Moreover, it is necessary to differentiate between TL in general and in specific settings [27,28]. In psychiatric care, in particular, two different levels of observation (psychiatric facility as a whole TL and specific elements within the psychiatric facility) must be considered [29].

### 1.3. TL and Facility Design in Providing MH Care

The clinical setting, especially inpatient psychiatric care, contributes substantially to MH [28,30]. Facility design, infrastructure, and architecture can both positively and negatively influence the health and recovery of service users (SUs) [30,31,32,33,34] and contribute to healthy workplaces [28]. Nevertheless, perceptions and assessments of environmental factors can differ between medical staff and SUs [30]. The United Nations Convention on Rights of Persons with Disabilities (CRPD), which provides a framework for human rights-oriented change to MH services [35], is closely linked to the improvement in the environmental quality in psychiatric care. The concept of TL can make an important contribution to this improvement.

Curtis et al. [28] identified six relevant TL dimensions, namely “Respect and empowerment for people with mental illness”, “Security an surveillance vs. freedom and openness”, “Territoriality, privacy, refuge, and social interactions”, “Homeliness and contact with nature”, “Places for expression and reaffirmation of identity, autonomy and SUs choice”, and “Integration into sustainable communities”, amongst different groups of people related to an inpatient MH facility (cf. 591). These dimensions include: (1) respect for and empowerment of people with mental illness and the extent to which the environment in a MH facility respects the personality, preferences, culture, and religion of SUs; (2) the conflict between the need to control and restrict SUs and the goal of promoting human needs and individuality; (3) the need within a MH facility for dedicated spaces (e.g., for social interactions) without coming into contact with medical staff; (4) a homely atmosphere and the use of natural elements to promote contact with nature; (5) the need to provide facilities that promote the MH of SUs while respecting individuality and diversity and enabling self-directed living and participation in treatment decisions; and (6) the promotion of (social) reintegration of people with mental illness through good networking of MH care facilities with the community environment [28].

However, despite the high relevance of the concept of TL in psychiatric care, still more research in this specific setting is necessary.

### 1.4. Aim of the Qualitative Meta-Analysis

The aim of this qualitative meta-analysis is to provide an evidence base that (1) summarises healing elements of TL in psychiatric facilities, (2) can inform future research and decision makers, and (3) differentiates the concept of TL from other concepts and definitions within psychiatric care.

Therefore, a systematic literature search and thematic analysis was conducted at the intersection of different scientific disciplines (such as geography, architecture, landscape ecology, landscape planning, environmental psychology, public health, and social psychiatry) and TL is summarily defined for the purpose of this qualitative meta-analysis as:

Landscapes where the symbolic environment [3], and “the physical and built environments, social conditions and human perceptions combine to produce an atmosphere which is conducive to healing” [24] (p. 96) as they contribute to physical health and MH in places [9,23].

## 2. Materials and Methods

Following the Population, Intervention, Control, and Outcomes (PICO) guidelines, initial searches were conducted until March 2021 using the search terms presented in Table 1.

Electronic searches were carried out using the four databases Medline (PubMed), PsycInfo, Cumulative Index to Nursing and Allied Health Literature (CINAHL), and Web of Science (Expanded). The respective search strategies can be seen in the electronic Appendix A. The inclusion and exclusion criteria presented in Table 2 were used to determine the papers to include.

This qualitative meta-analysis focused on sources published since 2000, because previously conducted reviews on the topic of TL [30,32,37] cover the initial research since the 1970s and state that environmental factors in healthcare settings can affect SU outcomes. Environmental stimuli can also be (part of) a specific medical treatment, as in light therapy for SUs with seasonal depression [32], or interventions such as forest or garden therapy. To examine the effects of environmental factors in healthcare settings and exclude therapeutic environmental stimuli interactions, studies in which environmental stimuli were applied as treatment or therapeutic interventions were excluded from this review. Study selection was an iterative process involving several stages (Figure 1). The process involved screening of titles and abstracts for relevance, and full text screening by H-L.S., L.O., T.P.L., and Z.K. In case of dissent, another person (T.M.) was consulted. Results are presented in accordance with the PRISMA 2020 explanation and elaboration: updated guidance and exemplars for reporting systematic reviews (PRISMA) [36] and the Qualitative Meta-Analysis Article Reporting Standards (QMARS) [38].

The data analysis was based on the TL framework [3,9] and was conducted by L.O., H-L.S., T.M., and T.P.L. The results of the included studies were extracted independently and used as units of analysis. In a first step, L.O. and T.M. categorised the results into physical, social, and symbolic TL dimensions. Then, L.O. followed an inductive approach and grouped similar findings into subthemes. This initial code system was discussed and modified by H-L.S., L.O., T.M., and T.P.L. Next, T.P.L. used this system to analyse the extracted results. Finally, H-L.S. examined whether any aspects were missing from an ecological perspective. No new themes emerged and it was decided that all relevant aspects were represented. The themes are illustrated in Figure 2. No specialised analytic software was used.

To assess the methodological quality of the included studies, parts of the Reporting Standards for Qualitative Research (JARS-Qual) by Levitt et al. [38] were used as an assessment scheme. The JARS-Qual include guidelines about which information should be reported from qualitative studies. The guidelines for the “method” and “results” sections of the JARS-Qual were used to record whether relevant information was provided. All studies were assessed regarding the six domains: (1) research design overview, (2) study participants or data sources, (3) researcher characteristics, (4) participant recruitment, (5) data collection, and (6) analysis. Specifically, we assessed whether all relevant information was provided (1 = high quality), information was provided partially or inadequately (0 = medium quality), or relevant information was missing (−1 = low quality). Each domain consisted of two subdomains with different quality indices and different numbers of indices per subdomain. The first three studies were independently assessed by L.O., T.P.L., and T.M. After these three studies, the feasibility and comprehensibility of the adapted JARS-Qual was tested and slightly modified. Inconsistencies within the results were discussed using specific quotes of the respective studies. Subsequently, the remaining studies were assessed by L.O. and T.P.L. and the results were consented. For each subdomain and domain, two characteristic scores (lower and upper 25% of the achievable score, e.g., −6 to 6: −3 ≤ −1; −3 > 0 ≤ 3; 3 > 1) were used to assess whether a study domain was of high quality (>75%), medium quality (25% to 75%), or low quality (<25%).

## 3. Results

### 3.1. Search Results

The database search yielded a total of 2853 studies, of which 16 duplicates were removed. After title/abstract screening, 2587 studies were excluded. Full text screening for the remaining 250 studies resulted in 13 included studies (Figure 1).

The characteristics of the included studies are presented in Table 3.

The reasons for exclusion are displayed in Table 4.

#### 3.1.1. Study Designs

The majority of the studies were qualitative, conducting interviews, and/or focus groups [39,40,41,44,47,48,51]. Six studies used a mixed-methods approach [42,43,45,46,49,50].

#### 3.1.2. Sample Sizes

Studies reported sample sizes ranging from nine to 114 participants. According to the reported sample sizes, data of 383 participants were analysed in the present qualitative meta-analysis. Five studies reported data for SUs only [39,40,41,43,44], two reported data on staff’s perspective on SUs [45,46], and one for carers [51]. Four studies reported data for more than one stakeholder group [42,48,49,50]. The two studies including quantitative data were not guided by power calculations.

#### 3.1.3. Study Population

Three of the 13 studies reported data on diagnostic characteristics of SUs. In the study by Agrest et al. [39], seven SUs were diagnosed with schizophrenia and three with other psychotic disorders, six SUs were diagnosed with a mood disorder or personality disorder, respectively, and two SUs were diagnosed with substance abuse. Hung et al. [42] reported data on geriatric SUs with depression, dementia, or both, with responsive behaviours. Within the three sub-samples investigated by McGonagle and Allan [43], the majority (67% of the bungalow SUs, 70% of the hospital ward SUs, and 100% of the community unit SUs) were diagnosed with schizophrenia. Two SUs in the study by Wood et al. [50] were discharged and 15 were forensic SUs. One study reported data on SUs who had attempted to abscond or succeeded in absconding [44]. The clinical staff members of one study, who reflected on physical design impacts on SUs, were associates of a mood and anxiety programme or a substance use programme [46].

The age of the SUs ranged between 19 and 72 years (Agrest et al. [39], mean age 34.1 years; Gilburt et al. [41], range 25 to 60 years; McGonagle and Allan [43], mean age of the bungalow SUs 55.0 years, hospital ward SUs 48.0 years, and of the community unit SUs 31.5 years) and for geriatric SUs the age range was 70 to 92 [42]. The mean gender ratio of SUs was rather balanced (overall 52.2% female; Agrest et al. [39], 54.0% female; Donald et al. [40], 55.0% female; Gilburt et al. [41], 47.4% female; Hung et al. [42], 42.9% female; McGonagle and Allan [43], bungalow SUs 61.1% female, hospital ward SUs 37.8% female, community unit SUs 54.5% female; Muir-Cochrane et al. [44], 67.7% female; Wood et al. [50], discharged SUs 50% female), except for the forensic subsample (15.4% female) in the study by Wood et al. [50]. McGonagle and Allan [43] reported mean lengths of the current stay in hospital for bungalow SUs (19 years, range 4 to 41) and hospital ward SUs (19 years, range 5 to 43). Seven of the nine carers interviewed by Wood et al. [51] were part of a carer group.

Staff members age ranged between 25 and 65 years, and years in profession within general psychiatric care ranged between one and 30 years in the study by Schröder and Ahlström [47]. The sample of the study by Wood et al. [50] comprised eight acute ward staff members (five female, three male), 16 forensic ward members (11 female, five male), senior staff member (one male), and 12 volunteers of a habitation exercise (eight female, four male). Two studies did not provide demographic and/or diagnostic characteristics of their study populations [45,49]. The study that investigated other relevant stakeholders not associated with daily psychiatric care, such as architects/designers or hospital managers, reported no descriptive characteristics of these participants [48].

#### 3.1.4. Setting

One study took place in Argentina [39], one in Canada [46], one in Australia [40], and one in both the United States of America and Australia [48]. Four studies were conducted in the United Kingdom (UK) [41,43,50,51], one in Sweden [47] and one in Denmark [49].

Two studies were conducted at psychiatric hospitals [41,49], one at a MH and substance use treatment facility [46], one at a mental and behavioural health facility [48], and one at a psychiatric day hospital [39]. Five studies recruited at psychiatric wards (two wards, each with a Low and a High Dependency Ward [40]; geriatric psychiatry ward [42]; acute psychiatric wards [44,45]). Three studies took place at more than one psychiatric setting (purpose-built bungalows, hospital ward unit, community units (hostels and associated flats) [43], psychiatric hospitals, and a psychiatric ward of a general hospital [50,51]. Schröder and Ahlström [47] recruited staff employed within general psychiatric care.

### 3.2. Methodological Quality of the Included Studies

Except for the one quantitative study [43] and the quantitative part of the study by Nanda et al. [45], all studies were assessed regarding the six domains and their respective subdomains of the JARS-Qual, as described in the method section. Overall, the quality of the assessed studies was low to medium. In terms of the research design overview, which allows other researchers to understand the rationale, method, and analysis of a chosen design and reach a similar conclusion about the results presented, only two studies provided sufficient information. Regarding the description of study participants or data sources, most authors provided adequate information on the number of participants or events investigated, but discrepancies were found in two studies. Demographic characteristics or other relevant information (e.g., diagnosis of SUs investigated) of the sample were mostly inadequate or missing. Information was missing in the areas of researcher characteristics and participant recruitment. The authors of the respective studies provided little information about their disciplinary background and whether there were researcher–participant relationships that might have influenced their research. The recruitment and selection process of participants was also insufficiently described. Overall, the data collection process was rated as medium quality because all included studies provided partially or inadequately sufficient information on the form of data collection and, except for one study, on the recording and transformation processes of their collected data, but no study indicated whether a data-collection protocol was used. The rigour and transparency of the analysis strategies showed deficiencies on average in “following the process completely”, but the “methodological integrity” subdomain was predominantly rated as transparent. The results of the quality assessment are presented in Table 5; the description and assessment scheme of used domains and subdomains are available upon request. The methodological quality of the quantitative study parts was not assessed.

### 3.3. Therapeutic Landscapes in Psychiatric Care Facilities

The results of the analysis based on the coding system (Figure 2) are described below. Note that the themes presented separately are interrelated and therefore some overlapping of aspects could not be avoided. The coding system with quotes from the original studies is presented in the electronic Appendix A.

#### 3.3.1. Physical Dimension

##### Design Features

The physical dimension comprises, on a descriptive level, the regular design of the facility, such as furniture or spatial layout.

##### Amenity

Amenity refers to the physical appearance of the facility. A sparse environment, e.g., poorly furnished or without distractions, is considered to impair SUs’ involvement in their healing processes. Poor physical conditions of the facility included lack of basic hygiene, housing comfort, availability of staff, and overcrowding [41]. Furthermore, lack of options was associated with boredom and prevented SUs’ distraction and relaxation, as did an existing but closed courtyard, which at the same time led to a feeling of being locked in [40]. In contrast, personal space, aesthetics and a calm atmosphere were relevant factors for SUs’ well-being [47]. A pleasant atmosphere may be easier to realise in small care units with SUs who have similar diagnoses [47].

For informal carers (e.g., relatives and friends), amenity is related to the accessibility and orientation in the facility, responsiveness of staff, and private places for visiting their cared-for relatives. Restrictors included access to the facility if the distance to transport was quite far and/or made access difficult for people with disabilities, and orientation within a facility [51]. In this regard, a staffed reception is as important as good signage or maps [51].

##### Space

This theme describes rooms and spaciousness for interactions between SUs with other SUs, informal carers, staff, and for group-based interventions. The natural environment was mentioned as a relevant aspect. The patio was described as a place to connect with others and the environment [42], and garden areas created a relaxing and restorative environment for SUs and their informal carers [51]. Although places where activities can be carried out promoted social engagement and helped to uphold SUs’ personality [42], private places were important for family contact and community life on the facility site [51]. Unfavourable were high noise levels in communal areas, which caused stress and were perceived as unpleasant [51]. To counteract this, both SUs and staff requested visiting rooms that also provide a place for one-on-one conversations. Apart from their designated purpose, quiet or calm rooms were used as visiting rooms for informal carers on the investigated acute wards. Another type of social space within the facility was a multi-faith room for religious purposes, neutrally designed to be symbolically inclusive for diverse communities [51].

Moreover, space affects the SU–staff relationship and the working conditions of staff. Staff reported that lack of working space impeded confidentiality during one-on-one encounters with SUs and was also time consuming [46]. Additionally, the therapeutic relationship itself can be compromised if the staff workspace is limited. The layout of the ward facilitated informal communication between SUs and staff, but impeded formal SU–staff interaction. From the staff’s perspective, this lifelike design was perceived as beneficial to the SUs, but contradicted the staff’s work requirements. Furthermore, the staff-designated room on the units was overcrowded and lacked space for other tasks, impeding the ability to act as role model in pro-social behaviour and restoration from the demanding clinical situation. Conversely, shared clinical rooms enhanced communication, professional collaboration, and quality of care [46].

##### Economic Benefits

Economic benefits may relate to cost savings in the spatial design of the facility. Art display versus no art display in an acute care facility showed that the cost of pro re nata medication per incident could be significantly reduced and entailed a cost saving of 60% when nature art was displayed [45].

##### Perception of the Physical Environment

In contrast to the descriptive level of the physical dimension, the perception of the physical environment comprises feelings associated with the spatial design of the facility.

##### Comfort

A comfortable environment creates homeliness and the feeling of being welcome. The physical environment of the facility thereby plays an important role. A homely environment, and (warm) coloured walls, comfortable furniture, and domestic decoration (such as paintings), were perceived as relevant components for healing and the therapeutic process [39,42,44,47], and elements such as fresh air, flowers, plants, and sunlight have also been reported to have a calming and refreshing effect on SUs [42]. In addition, being able to welcome family members [39], and an overall tranquil and calm environment [42], promoted SUs’ MH. Equally important to a “safe, caring, and comfortable” environment is staff [40] (p. 65). In contrast to comfort, long and straight corridors, wheelchairs left on site [42], noise (which was associated with feelings of anxiety, distress, and helplessness [42,44]), unfamiliarity, overcrowding, hustle, ugliness (prison-like), and uncomfortable temperatures [44] were reported. Similarly, informal carers reported the restricted use of rooms (e.g., bedrooms) to meet their cared-for relatives and lack of facility comfort as unpleasant [51].

Within an open and transparent facility, concept staff can unobtrusively monitor SUs [49]. However, SUs were able to observe staff simultaneously, which was perceived as uncomfortable, intimidating, and unethical. Furthermore, the design interfered with staff’s need for privacy and their ability to handle daily work activities (e.g., preparing injections).

##### Staffs and Stakeholders View on SU Outcome

This theme comprises secondary perspectives on the impact of the physical environment on SU outcomes. Regarding different types of artwork, the staff interviewed reported observing only minor differences in SUs’ responses between abstract and representational art [45]. The art was predominantly viewed by SUs without verbal or physical reaction. Overall, staff rated the artwork as a positive feature of the physical environment, particularly realistic nature art (e.g., calming), as being better and more beneficial for SUs than abstract artwork (e.g., irritating, ugly, headache inducing). However, staff were concerned about possible negative effects of abstract art on already psychotic SUs and recommended paintings with socially engaging content.

From staff’s perspective, the architectural design of lifelike facility units through private bedrooms and the aesthetic design led to a strengthened sense of freedom, autonomy, and independence in SUs, in addition to being conducive to recovery by enabling transition into the community [46]. The findings of Shepley et al. [48] only touch on potentially important environmental aspects, but do not describe them in detail. According to the respondents, a facility should be deinstitutionalised and homely. For furniture, the requirements need to be balanced between attractiveness, non-institutional designs, and safety aspects. Less orderly and organised environments contribute to more autonomy and spontaneity of recovery. Private bedrooms and bathrooms were preferred over shared rooms to provide privacy and a retreat, and to normalise the stay in the facility. The degree of privacy offered should depend on the individual SU and their diagnosis. Furthermore, respondents reported a need for places to socialise and to encourage SU–staff interactions. Flexible and mixed seating should allow for spontaneous rearrangement of the environment. Day rooms, communal areas, multipurpose rooms, and outside areas (e.g., gardens) were stated as being equally relevant for SU–staff interaction and therapy. In particular, visual, and physical access to nature and daylight was named as important.

#### 3.3.2. Social Dimension

##### Features of the Social Dimension

The social dimension includes the interaction between the individual and the environment, aspects, and ambiguities related to the purpose of places, SU admission, personal freedom, and social connections.

##### Confusing Space

This theme involves the description of spaces, associated with uncertainties about interactions in and with these spaces, and the spaces’ purpose. A perceived discrepancy between a sterile psychiatric environment in need of comfort and a reassuring and safe atmosphere can lead to confusion for Sus; for example, treatment rooms with floor-to-ceiling glass were perceived as diametrically opposed to privacy [40]. Moreover, ambivalent encounters with different people, such as support and efforts by staff to create a pleasant environment, or violence by other SUs or security staff, contributed to confusion.

##### Safe Space

Psychiatric facilities need to provide a safe and secure environment. The findings include aspects of admission, the stay in the facility, and specific design features that facilitate or hinder perceptions of safety.

Admission to a facility is highly sensitive [44]. The mental illness and the environment influenced the perceived safety in the facility. Feelings of panic, fear, and confusion, associated with admission and medication, can lead to absconding.

In terms of the facility design, the level of privacy varied and was perceived differently by SUs [44]. Striking a balance between privacy and security can be difficult. On the one hand, privacy was considered as a positive aspect and the facility was perceived as safe if it provided a sanctuary and protection. However, “too much” privacy can make the facility feel less safe because SU can abscond or gain access to strangers. Similarly, the perception of the facility environment plays a role. Unfamiliarity was associated with discomfort and insecurity, whereas familiarity can provide a sense of security [44]. Furthermore, social relationships can be influenced by the physical environment, i.e., women reported feeling less safe in communal areas with men [44]. Additionally, feelings of personal safety and security can be supportive factors in the therapeutic process and may reduce distress [42]. Particularly for the safety of elderly SUs, fall prevention is needed, e.g., through the availability of handrails or benches [42].

In addition, opportunities for retreat and safety were crucial for staff [48]. Private rooms in particular can pose dangerous situations for staff. Therefore, a balance should be maintained between open (e.g., SU visibility) and closed (e.g., safety concerns, barrier for SU–staff relationships) design of nursing stations.

##### Encouraging/Discouraging Space

The facility design must encourage SUs’ autonomy, so that they can move freely within the facility, and make independent decisions about daily tasks and social activities.

Physical freedom is a basic human right and its lack was associated with mental distress and a prison-like perception [41]. Physical freedom comprised the ability to move freely inside and outside the facility, and to connect with the (natural) environment, but there were inconsistent exit regulations between the facilities. Although leaving the premises was allowed in some facilities, in others it was prohibited. A perception of the psychiatric environment as prison-like was associated with feelings of confinement, subordination, and dependency [44]. A lack of choice and structured activities can lead to loneliness, isolation, and boredom, and hinder the facility from functioning as a TL.

Smoking regulations and monitoring of smoking differed on each ward, but regulations that interfere with the freedom of choice to smoke were particularly negatively perceived by both SUs and staff [50]. Patios that eliminate the need for escorts to smoking areas were associated with relaxation and personal freedom, whereas strict smoking regulations (including assigned breaks) were seen as a means of social control for SUs. In the latter, smoking was seen as a major event, sometimes leading to an increase in tobacco use. Therefore, tobacco addiction may not only have been significant for smoking behaviour, but also boredom, the feeling of being disempowered, and the fact that smoking areas were more conducive to relaxation than the ward. Moreover, this can lead to a disadvantage for non-smokers if the areas are not communal but reserved for smokers. Regardless, possible (health) effects of passive smoking can have a disruptive effect on the environment and lead to an increase in stress and anxiety among non-smokers. As staff was not allowed to smoke in assigned areas, finding secret smoking spots could be seen as an escape of institutional control or as a way to self-regulate. The different ways in which smoking behaviour was controlled show how social control was exercised depending on the role in the facility.

A supportive environment enables high levels of independence in daily tasks [42]. In particular, long corridors, identical-looking rooms, and frequent room changes interfered with the independent orientation of geriatric SUs. Lack of directional cues and signage exacerbated this problem. More openly designed units provided SU privacy and independence, including decisions about whereabouts in private bedrooms and communal areas, and were associated with improved quality of life and empowerment [46]. Moreover, the layout encouraged the development of relationships through clear boundaries and control of personal space in a salutary way.

Bungalow facilities (i.e., lifelike environments embedded in the community) were associated with significantly less moderate and severe social problem behaviours compared to traditional ward settings, but demanded more stable levels of domestic skills and psychological impairment [43].

Meaningful activities, personal growth, and interpersonal relationship can improve SUs’ MH [39]. The provision of activities led to feelings of being purposeful, contributing, capable, and normal [42], and the effect of music, in particular, induced positive emotions, humour, and memories [42]. In this context, SUs requested activities that enable nature connection, such as aroma-therapy, footbaths, and gardening [40].

##### Social Connection

The psychiatric environment should encourage and facilitate SU’s social relationships with other SUs, informal carers, and staff. Supportive elements (e.g., music) encouraged SUs to connect with others and actively use spaces [42]. SU–staff relationships were important for SU MH. When SU–staff relationships were deficient due to unavailability of nurses, SUs’ well-being decreased [40]. The location of the nursing station may also affect the SU–staff relationship if it causes a lack of interaction and communication and a sense of separation [44]. Furthermore, informal carer–staff relationships can enhance satisfaction with care [51]. Informal carers expressed a need for places that “create supportive social connections” [51] (p. 127). Limited communal areas were seen as discouraging social interaction and religious expression. Furthermore, continuously locked wards, where staff controlled every person entering and leaving the facility, undermined the position of informal carers and SUs in the facility. Moreover, the restricted visiting hours hindered SU–informal carer interaction and led to social isolation. In addition, remoteness of the facility led to a lack of community-based care options while burdening socioeconomically disadvantaged informal carers with travel time and costs.

Further, smoking areas were perceived as social spaces as they were used not only by smokers but also by non-smokers who sought to access the outside by socialising with smokers [50]. Additionally, allocated areas could facilitate the relationship between staff and SUs, as a sense of belonging is fostered in these areas [50].

#### 3.3.3. Symbolic Dimension

##### (Therapeutic) Value

The theme “value” comprises the attributed therapeutic value of the psychiatric care environment. It summarises the facilitators and barriers that affect whether a facility is perceived as therapeutic, i.e., whether it supports SUs’ recovery process, informal carers’ needs, and staff satisfaction with their work.

##### Facilitators

The physical and social dimension included elements such as personal space, privacy, and homeliness [39,40,42,44,46,47,48], and restorative environments [42], e.g., with design elements such as art [45] or natural elements inside [44], which have been found to enhance SUs’ improvement. In addition, places that provide activities and access to the natural environment [42,48,51], and areas to practise religion [51], have been mentioned as important for recovery. Furthermore, the provision of lifelike environments [43,46], autonomy [42,46,48,50], and freedom [41,44] were reported as relevant aspects. Overall, complaisant relationships between all stakeholders, i.e., SUs, informal carers, and staff [39,40,42,44,51], in addition to (meaningful) activities [39,40,42], were discussed as notable for SUs’ recovery.

##### Barriers

In contrast, a sparse environment that offered limited opportunities for distraction or relaxation [40,41], may be considered a barrier for SU improvement. Moreover, uncomfortable temperatures, overcrowding, hustle [44], and noise [42,44] have been found to be barriers to SU MH. Confusion occurred through uncertainty of admission [44], the purpose of spaces [40], and unfamiliarity of the environment [40,44]. Stigma by design has been found as a barrier [46,51]. Informal carers [51] and staff [46] described an old facility design as supporting the public’s misconception of mental illness. In contrast, newly redesigned units were expected to convey a more positive image of the facility and reduce stigma [46].

Doorway transitions have been identified as conflictual in two ways. First, locked, staff-controlled wards that regulated the transition of informal carers from outside to inside were perceived as undermining the position of informal carers and SUs [51]. Second, door encounters (transition to the nursing station) were critical for staff and SUs [49]. During these encounters, SUs often made appeals and requests, which sometimes disrupted staff activities. These interactions were often shifting from social to supply and demand interactions. Even when the door was closed only during times of meetings or activities requiring privacy, this led to frustrations among SUs, while requests made during these times led to annoyance among staff.

Just as for SU, privacy [48,49] and safety [48] were important to staff. In particular, the lack of designated rooms for staff and a more SU-focused design that conflicted with staff working conditions [46] reduced staff satisfaction. Furthermore, insufficient therapeutic space was a barrier for staff to fulfil their therapeutic tasks [46].

## 4. Discussion

The aim of this qualitative meta-analysis was to review MH effects of TL within psychiatric care facilities. Hence, the TL framework described by Gesler [9,24] was used. Perspectives of different stakeholders, e.g., SUs, informal carers, and staff, from various psychiatric facilities (e.g., psychiatric hospitals, specific wards, and day hospitals) were included and the results embedded in the three main categories: physical, social, and symbolic dimension.

Within the physical dimension, design features and their perception are included. A unanimous opinion was the need for a caring, homelike, and pleasant environment. In this regard, a balance between a pleasant and useful environment, evenly offering privacy and safety for SUs and staff, is necessary. Moreover, a facility should enable possibilities for reflection without disturbance but also positive distraction by the TL. Hackett et al. [52] found similar results regarding the physical environment for youth SUs in different healthcare settings. Service providers and youth SUs mentioned a welcoming environment, i.e., being bright, decorated, and youth-friendly, to be important for high quality care. Moreover, youth SUs asked for privacy and autonomy in decorating their own rooms. An open view and access to nature were mentioned as important aspects of the physical environment by some of the referred studies. Visual and physical access to nature have been found to be important for healthcare facilities in general, as being not only beneficial for the physical and mental health of SUs, but also that of staff [53]. In the same way, the provision of daylight within the facility enhances SUs’ and staff’s physical and mental health [53]. In the study by Shepley et al. [48], there was an awareness for the need for daylight, but no idea of realisation. According to Sherif et al. [54], daylight and the view from a window can be influenced and regulated by the shape of the slats on blinds. In this regard, the possibility of personal control of the individual’s environment can benefit MH additionally [53,55]. Moreover, special designs/furniture for certain SUs, e.g., geriatric or forensic SUs, are needed. According to Karlin and Zeiss [56], an issue that must be considered is the design of the interior in such a way that goals of stimulation are addressing the right SU, while fostering an optimistic sense about hospitalisation at the same time. It is desirable to design psychiatric facilities that align with different SU demands, but this may be not completely feasible in every case [33,56]. Liddicoat [57,58] describes the difficulties when SUs perceive traces of others or environmental cues, such as smells or fabrics, triggering traumata within therapeutic spaces, and highlighting the challenge to find a balance between over- or under-stimulating environments. Another important aspect is safety, in terms of suicide resistance, which can be achieved through the practical implementation of standardised guidelines for safe environments. In a study by Watts et al. [59], the suicide rate decreased by 62% after implementing the Mental Health Environment of Care Checklist. According to the review by Thibaut et al. [60], safety of the physical environment includes measures such as door locking or placement of SUs within MH facilities, which can be part of organisational management [61]. Although it can be assumed that a pleasant and welcoming environment is relevant for a wide range of SUs (e.g., different mental illnesses, age groups), specific requirements regarding the geographic location, the setting, and SUs must be considered.

The provision of rooms for social connection and activities with other SUs, carers, and staff, and to practice religion, is important and directly connected to a conducive social environment. This is in line with results from Jovanović et al. [62], who found that family rooms off ward were associated with psychiatric SUs’ treatment satisfaction. Most of the investigated wards in this study were part of general hospitals; hence, it may be easier to provide social places off-ward than at solely psychiatric facilities. Nonetheless, it is likely that the availability of specific places to meet family and friends, apart from communal rooms, enhances SU treatment satisfaction and thus well-being. In this regard, informal carers’ demands, such as accessibility to the facility and places to connect with their cared-for relatives, are essential and enhance social interactions [53]. Moreover, multiple-occupancy rooms can provide the opportunity for social interaction between SUs [63,64]. In addition, Ulrich et al. [65] mentions the stress-reducing effects of communal areas with movable seating and ample space, which allows the regulation of relationships. Providing unlocked outdoor gardens and rooms with a view of nature can foster stress reduction by offering pleasant places to seek privacy or socialise [65]. Karlin and Zeiss [56] recommend designing social places so that SUs can control their level of social contact, retreat, or form new relationships. The need for areas and (sportive) activities, where the mental illness is not present for SUs or visible for others, allowing them to adapt to roles apart from being a SU, was also outlined by McGrath and Reavey [66]. Regarding (meaningful) activities or areas, it is important to take SUs’ preferences into account. As shown by Parkinson et al. [67], activities, i.e., horticulture, are not meaningful per se. SUs perceived horticulture only as beneficial when related to the individual’s interest [67].

A relevant aspect of the social dimension is a balance between sense of safety, privacy, autonomy, and freedom, e.g., to allow everyday choices and prevent the feeling of being locked up. Similar to the results of this qualitative meta-analysis, SUs in several studies repeatedly expressed feeling locked-up and described the environment as prison-like [52,64,66]. Clear boundaries and options on how to use private and common places, e.g., no formal SU–staff interaction in private rooms, can be supportive for treatment, but need to fit SUs’ demands. Although in the present qualitative meta-analysis SUs expressed a desire for (great) leeway, Maloret and Scott [68] showed that specific SU-groups (SU with autism spectrum disorder and an acute panic disorder) need daily routines and structure. Single-SU rooms are associated with privacy [53] and autonomy through the sense of control of the SUs’ own environment [53,64]. A review of MH aspects of the built environment affirms the importance of indirect control of the built environment through social interactions, in a variety of settings [55]. In one included study, women felt less safe in communal spaces [44]. In contrast, Jovanović et al. [62] found that SU treatment satisfaction is associated with mixed-sex wards. Consequently, further research is needed as previous research stated both advantages and disadvantages of mixed-sex wards [69,70]. Regarding safe spaces, staff have to be considered [71,72], because about 24–80% of MH care staff experience violence at least once in their career [73].

From SUs’ perspectives, lifelike environments are preferred over restrictive ones, albeit staff’s demands to execute their working tasks need to be considered. This matches previous findings of a preference for lifelike units that are associated with beneficial MH outcomes [74]. Earlier-mentioned elements, such as daylight, music, and airflow [75], hospital gardens [76], and separate places for staff restoration and communication [53], can also enhance staff’s work satisfaction.

Concurrently, social relationships are of significant importance, as, e.g., SU–staff relationship can have a direct effect on SU well-being. The physical and social dimensions act as facilitators and barriers of the therapeutic value, i.e., symbolic dimension, of the psychiatric facility. An unsuitable physical environment can severely impair aspects of the social dimension, such as quality of care and the feeling of safety, and may damage the person-centred therapeutic process [64]. Additional barriers are the associated stigma through the facility design, in addition to SUs’ and carers’ subordinate position within the facility. This was also mentioned by McGrath and Reavey [66], where the entire environment and, in particular, locked doorways, reflected the commonly perceived stigmatisation and caused a feeling of devaluation.

This qualitative meta-analysis revealed a research gap regarding the effect of natural environments (e.g., green and blue spaces) in psychiatric care facilities. Many studies from diverse disciplines suggest benefits of nature for human health, and especially MH, but the evidence for psychiatric settings is lacking. The included studies only incidentally mention different aspects of contact with nature. Furthermore, the characterisation of the natural environment rarely goes beyond the dichotomy of natural or artificial. This has been criticised before, concerning health benefits of nature in other settings and target populations [4,20,77]. Studies that examined nature-based therapeutic approaches, e.g., horticultural therapy and greenspace interventions, were excluded here but have already been reviewed [78,79,80]. They concluded that nature-based approaches can be effective for MH but also found inadequate methodological quality of the included studies. Additionally, a reporting bias was detected, creating a knowledge gap regarding obstacles and neutral or negative experiences [78,80]. This underlines the need for transparent and methodologically sound research of the contribution of natural environments and elements to TL.

A positive aspect is that various stakeholders were involved in the included studies; however, this led partly to a diffusion of perspectives. It is necessary to differentiate whether a SU, carer, or a staff member mentioned a potential positive or negative aspect for their own or as an assumption about another target group. In particular, the intersection of SU needs and staff-related problems can be a conflict that needs to be addressed in MH facilities [81], and which is also affected by the staff’s control strategies and SU feelings of being monitored [82]. Nonetheless, it is important to find a consent between different stakeholders about beneficial and obstructive aspects of TL within the psychiatric care setting. In this regard, both carers and staff have a dual role. Carers have their own demands when interacting with and within the facility, but are also a voice for their treated relatives’ concerns. Similarly, staff have their own requirements for their working environment and satisfaction, but also want to meet SU needs. However, these demands can be diametrically opposed. To improve the evidence-based criteria for a beneficial TL in psychiatric care facilities and enhance the quality of care and human rights, the research gaps revealed by this qualitative meta-analysis should be assessed according to the European Psychiatric Association’s recommendations addressing a wide range of aspects of quality assurance in mental healthcare [83] and the QualityRights initiative launched in 2013 by the World Health Organization [84,85].

In comparison to former systematic reviews, certain similarities and differences can be demonstrated. Dijkstra et al. [32] investigated physical environmental stimuli that turn healthcare facilities into a healing environment through psychologically mediated effects. Although conclusive evidence is limited, three relevant dimensions of environmental stimuli were discriminated for health and well-being of SUs: ambient, architectural, and interior design features, which overlap with the results of this qualitative meta-analysis. However, the present qualitative meta-analysis concludes that the available evidence regarding environmental stimuli has not substantially improved within the last 15 years. In addition, Papoulias et al. [86] studied the effects of ward design on SU symptoms and clinical outcomes, treatment satisfaction, perception of ward atmosphere, well-being, and behaviour of SU and staff since the 1970s. Several of the included studies found that design elements of the physical environment were associated with non-clinical outcomes, but found no strong causal link to clinical outcomes. This qualitative meta-analysis also reveals extensive methodological limitations and the need for quantitative and novel methods, but adds to the requested qualitative studies. Dushkova and Ignatieva [3] investigated the positive impact of nature on human physical and mental health. A key finding was the representation of the different elements of TL on an equal structural level, whereas this qualitative meta-analysis found a rather hierarchical structure of the dimensions of TL. Regardless of the different methodological approaches/findings, similar conclusions can be made for the connection between the quality of different landscape elements and MH. The impact of physical environment, art, and design in mental healthcare was explored by Daykin et al. [30]. Similar to this qualitative meta-analysis, they found that arts may have positive effects on clinical (e.g., reduction in anxiety and depression) and behavioural outcomes, especially when containing calming naturalistic and domestic imagery [30]. Moreover, the diversity of views and perceptions included important differences between staff and SUs, and between different SUs. In addition, they identified positive effects, such as reduced stress, reduced risk, and improved way-finding, and enhanced perceptions of healthcare environments. In contrast, no social or symbolic dimensional aspects were addressed. Van den Berg [37] summarised clinical and psychological effects of nature, daylight, fresh air, and quiet in healthcare settings. In contrast to the present qualitative meta-analysis, the review made strong conclusions, that there is sufficient evidence that viewing nature can reduce stress and pain, and weak evidence for the presence of indoor plants to have positive effects on mood or physical discomfort, and daylight in buildings to have beneficial health effects. In their systematic review, Taheri et al. [27] evaluated any type of TL with a focus on health dimensions. Corresponding to the results of this qualitative meta-analysis, they concluded that TL (e.g., in psychiatric care facilities) have the potential to promote recovery as long as they take into account the needs and characteristics of the SUs.

### Limitations

The explicit aim of this qualitative meta-analysis was to gain a broader insight into MH effects of TL within psychiatric care facilities. Transparency was provided through compliance with reporting standards and the quality assessment of the studies. Notably, subjectivity is not entirely excludable in this process as it “involves the interpretive aggregation of thematic findings” [38] (p. 40) and is worthy of discussion. The results were heterogeneous in terms of the settings and the investigated participants. We state that we synthesised these results meaningfully into broad themes, but nonetheless evidence for most outcomes is missing. Hence, the quality of the evidence must be rated as insufficient.

As described before, this qualitative meta-analysis revealed a gap regarding natural environments in the psychiatric care setting. We explicitly included search terms regarding the natural environment, but excluded studies that directly investigated therapeutic purposes, such as garden therapy. It cannot be ruled out that, through this strict approach, possibly relevant aspects were disregarded.

## 5. Conclusions

This qualitative meta-analysis summarises how different stakeholders, e.g., SUs and staff, perceive certain aspects within psychiatric care facilities as supportive or unsupportive of SUs’ improvement in MH, and for carers’ and staffs’ (working) satisfaction. It outlines that environmental factors (i.e., aspects of the physical, social, symbolic dimensions) can have a positive and a negative impact on SU MH and recovery. Furthermore, demands of different stakeholders need to be evenly considered when psychiatric facilities are to be (re)designed. In this regard, the concept of TL is useful. In addition, the results may be relevant for decision makers (e.g., for health facility design) as they show that other aspects of design (e.g., a caring, homely, and comfortable environment) are important for the needs and health of SUs, in addition to clinical outcomes and clinical functionality. These should be consistent with evidence-based criteria for improving quality of care and human rights in MH.

However, the results of the present qualitative meta-analysis lead to several questions. These include, but are not limited to: (1) is the structure of the main dimensions of TL equal or rather hierarchical in this specific setting, (2) which specific design features (evidence-based design), physical and organisational, must be considered, (3) how do specific groups differ, e.g., gender, age, or various disorders, in terms of their needs and how these can be met, (4) to what extent do elements of the natural environment contribute to TL in psychiatric facilities, and (5) which evidence-based practical recommendation can be given to decision makers in the future? To answer those questions, in particular, quantitative research designs that investigate the relationship between the physical environment and (mental) health outcomes, are necessary. In this respect, single environmental stimuli need to be further researched. Moreover, practical implications of elements, which can be applicable in general or need to address specific demands, are needed. To contribute to these knowledge gaps, the transdisciplinary junior research group “Healthy Places—Therapeutic Landscapes (LebensLand)” will focus on some of these aspects.

## Figures and Tables

**Figure 1 ijerph-19-01490-f001:**
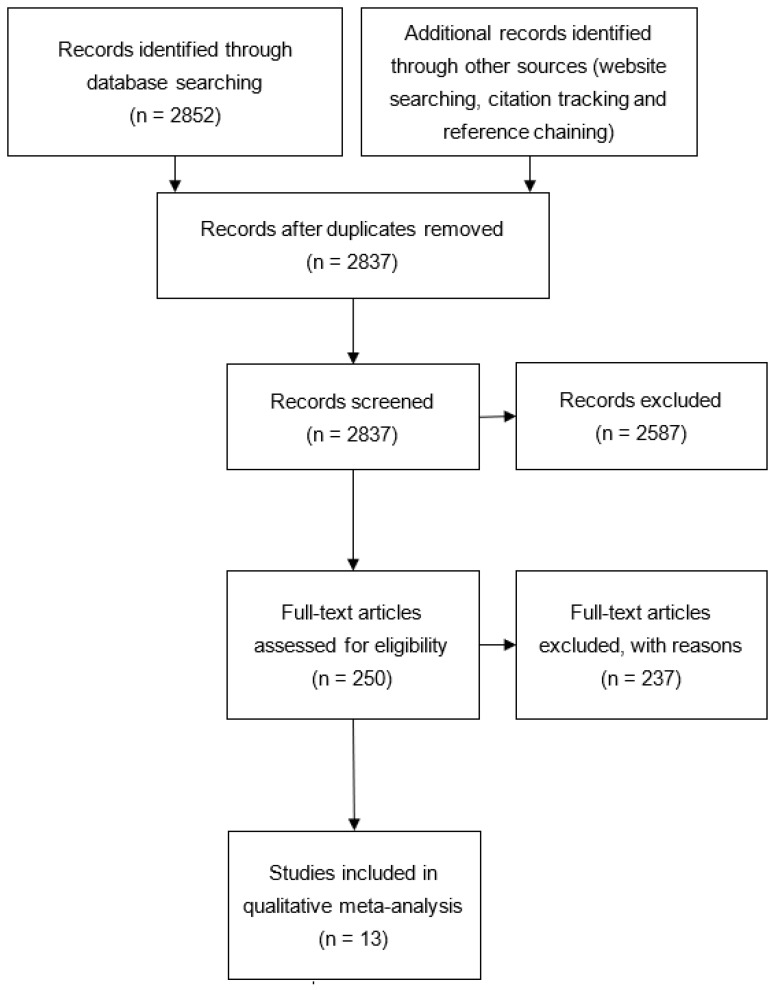
PRISMA flow diagram.

**Figure 2 ijerph-19-01490-f002:**
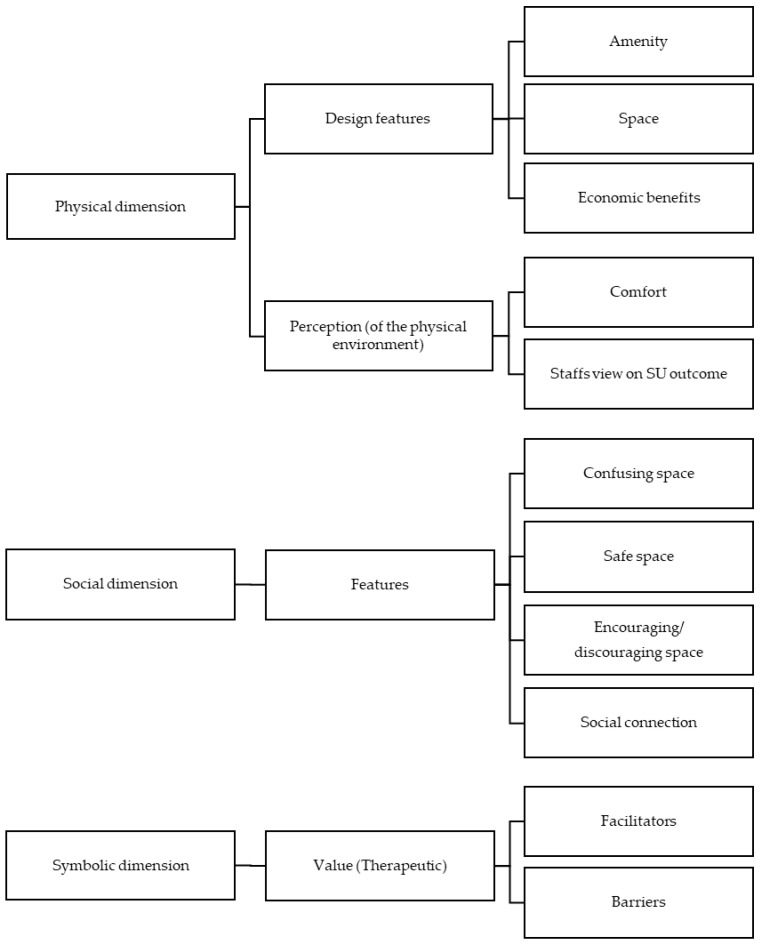
Coding system.

**Table 1 ijerph-19-01490-t001:** Search terms used in the electronic search using PICO [36].

Population		Intervention
Word Group 1	Word Group 2	Word Group 3
Mental Health ^a,b^	Health Facility Environment ^a,b,c^	Therapeutic Landscapes
Mental wellbeing	Mental health service ^b,c^	Therapeutic assemblage
Mental Health Rehabilitation	Hospitals ^b,c^	Gardening ^a,b^
Mental disorders ^a,b,c^		Therapeutic mobilities
Stress ^c^		Ecosystem services
Mental health care		Nature-based solutions
Neurological Rehabilitation ^a,c^		Healing Gardens
Psychiatric Rehabilitation ^a,c^		Green care
Psychological distress ^a,c^		Streetscape
		Green space
		Blue Space
		Landscapes
		Environment ^a,b,c^
		Virtual environment
		Horticulture ^a,b,c^
		Natural Resources ^a,c^
		Neighbourhood
		Architecture ^a,b,c^
		Healing Environment
		Built environment ^a,b^

Notes. Terms within word groups combined using “or”; word groups combined using “and” Subject Heading-Terms: ^a^ Medline (PubMed) ^b^ CINAHL ^c^ PsycInfo/PsyIndex.

**Table 2 ijerph-19-01490-t002:** Inclusion and exclusion criteria.

Inclusion Criteria	Exclusion Criteria
Interventions that investigate the health effects of environmental stimuli in the mental health care setting	Studies in which interventions are part of the therapy
Period: Studies from 2000 on	Simulation Studies
Adults over 18 years	Studies that address environmental pollution
Language: German and English	Studies where the therapeutic landscapes were confounded with non-environmental changes, such as changes in the nursing care policy
Industrialised Countries	Any type of reviews (e.g., systematic review, scoping review, rapid review, literature review)
Outcomes of Interest: Mental health Outcome	
Target groups: service user and medical staff	
Service user staying in a healthcare setting for any length of time	

**Table 3 ijerph-19-01490-t003:** Characteristics of the studies.

**Reference**	**Agrest et al. [39]**	**Donald et al. [40]**	**Gilburt et al. [41]**
Disciplinary affiliation of the primary author	Proyecto Suma, Community Mental Health Service, Buenos Aires, Argentina	School of Psychological Sciences, Monash University, Melbourne, Australia	Health Service and Population Research Department, Institute of Psychiatry, Kings College London
Geographic location	Bueno Aires, Argentina	Melbourne, Australia	England
Language	English	English	English
Method of data collection	Two stages (1) Interviews (2) Focus groups	Semi-structured interviews, focus groups	Interviews/Focus groups
Method of analysis	Grounded theory	No methodological background stated	Thematic analysis
Setting	Day hospital	Psychiatric department of a metropolitan hospital	Service user who had each experienced admission to a psychiatric hospital in England
Study Period	Not specified	Not specified	Not specified
Sample	Service user (*n* = 24)	Inpatients (*n* = 20)	Service user (*n* = 19)
Recruitment method	(1) Purposeful intensity sampling (2) “Researchers presented the project to users at the day hospital assembly where users could choose to participate in the focus groups. No selection was used.” (Agrest et al. [39] (p. 3))	“(…) direct approach by a member of the research team.” (Primary reference, p. 64)	Volunteer sampling (mental health resource centres as recruitment sites; advert in a local mental health charity newsletter
Purpose of the study	“(…) identify the elements of day hospital treatment that facilitate or hinder users’ recovery process within a day hospital in Buenos Aires, Argentina.” (Agrest et al. [39] (p. 2))	“(…) secure deeper understandings (…) of how participants perceive or give meaning to their illness experience, and the receipt of care in psychiatric settings.” (Donald et al. [40] (p. 64))	“(…) explore the experiences of admission to acute psychiatric hospital from the perspective of services users.” (Gilburt et al. [41] (p. 2))
**Reference**	**Hung et al. [42]**	**McGonagle and Allan [43]**	**Muir-Cochrane et al. [44]**
Disciplinary affiliation of the primary author	School of Nursing, University of British Columbia, Vancouver, Canada	Stafford Centre, Derbyshire Mental Health Services (NHS) Trust, Kingsway Hospital Derby	Candice Oster, School of Nursing and Midwifery, Flinders University of South Australia
Geographic location	NA	United Kingdom	NA
Language	English	English	English
Method of data collection	Observations; “go-along” interviews	Cross-sectional survey; patient record data	Interviews
Method of analysis	Thematic analysis	Kruskal-Wallis one-way analysis of variance; Mann-Whitney U-tests	Thematic analysis
Setting	Geriatric psychiatry unit	Purpose-built bungalows; hospital ward; community units (hostels and associated flats)	Acute psychiatric wards (open/closed wards)
Study Period	Not specified	Not specified	Not specified
Sample	Geriatric patients (*n* = 7), family members (*n* = 4)	Residents (*n* = 66)	Inpatients (*n* = 12)
Recruitment method	Purposive sampling	“All residents within the RCCS formed the population under review.” (Primary reference, p. 495)	Purposeful sampling
Purpose of the study	“(…) how the physical environment in a geriatric psychiatry unit may play a role in either supporting or obstructing patient and family care needs.” (Hung et al. [42] (p. 2))	Compare level of social behaviour problems between three different resident groups	“(…) explore the experiences of people who had been held involuntarily under the local mental health act in an Australian inpatient psychiatric unit, and who had absconded or attempted to abscond”. (Muir-Cochrane et al. [44] (p. 306))
**Reference**	**Nanda et al. [45]**	**Novotná et al. [46]**	**Schröder and Ahlström [47]**
Disciplinary affiliation of the primary author	Vice President, Director of Research, American Art Resources, Houston	Department of Psychiatry & Behavioral Neurosciences, McMaster University, McMaster Children’s Hospital	Psychiatric Research Centre, Primary Care, Psychiatry and Rehabilitation, Örebro County Council; Department of Medicine and Care, Division of Nursing Science, Faculty of Health Sciences, Linköping University
Geographic location	NA	Ontario, Canada	Sweden
Language	English	English	English
Method of data collection	Mixed-Method Design (Interviews; Pro re nata (PRN)-medication data)	Focus groups; Observations	Semi-structured Interviews
Method of analysis	Thematic analysis; unpaired *t*-test; efficiency analysis	Content analysis	Phenomenographic analysis
Setting	Multi-purpose lounge of an acute care psychiatric unit	Mental health and substance use treatment facility	Psychiatric Care
Study Period	Not specified	2007–2009	Not specified
Sample	Nurses (*n* = 22)	Staff members (*n* = 30)	Care staff (*n* = 20)
Recruitment method	Not specified	Convenience sampling	“The subjects who were chosen by the supervisor at the particular work-place received a letter about the study.” (Primary reference, p. 205)
Purpose of the study	“The objective of this study was twofold: to lay the foundation for the use of art in mental health facilities to reduce patient anxiety and agitation, and investigate of the economical ramifications of this impact on the healthcare organization.” (Nanda et al. [45] (p. 388))	Impact of the Physical Design on (a) Clients; (b) Service Delivery; (c) Work Environment	“(…) describe how the psychiatric care staff and care associates perceived the concept of quality of care in the case of psychiatric care.” (Schröder and Ahlström [47] (p. 204))
**Reference**	**Shepley et al. [48]**	**Simonsen and Duff [49]**	**Wood et al. [50]**
Disciplinary affiliation of the primary author	College of Human Ecology, Cornell University, Ithaca, NY	Department of Business IT, IT University Copenhagen, Copenhagen, Denmark	Department of Geography, Durham University, UK
Geographic location	United States; Australia	Slagelse, Denmark	Northern England
Language	English	English	English
Method of data collection	Interviews, Focus groups	Semi-structured qualitative interviews; Observations	Interviews, Group discussions; Observation
Method of analysis	Grounded theory		Thematic analysis
Setting	Mental and Behavioral Health Facilities	Psychiatric hospital	New Hospital compared with the older facilities it replaced
Study Period	Not specified	Not specified	2010–2011
Sample	Interviewees (*n* = 19): clinicians (*n* = 7), academics/researchers (*n* = 4), architects/designers (*n* = 5), researcher/practitioner (*n* = 1), administrators (*n* = 2)	Hospital associates (*n* = 17): head-physician (*n* = 1), head-nurses (*n* = 3), nurses (*n* = 5), auxiliary nurses (*n* = 3), care worker (*n* = 1), hospital managers (*n* = 2)	Participants (*n* = 114): service user (*n* = 17), carer (*n* = 1), staff members (*n* = 25), volunteers of a habitation exercise (*n* = 12)
Recruitment method	Snowball sampling	NA	Not specified
Purpose of the study	“(…) identify features in the physical environment that are believed to positively impact staff and patients in psychiatric environments and use these features as the foundation for future research regarding the design of mental and behavioral health facilities.” (Shepley et al. [48] (p. 15))		“(…) perceived significance of spaces used for smoking, and their importance for wellbeing of patients, staff and others using the hospital buildings. We discuss how this is associated with the socio-geographical power relations that influence smoking behaviour and how our findings contribute to theorisation and practical application of ideas about therapeutic landscapes.” (Wood et al. [50] (p. 8))
**Reference**	**Wood et al. [51]**		
Disciplinary affiliation of the primary author	Durham University, Wolfson Research Institute, Queens Campus; Department of Geography, Durham University, UK		
Geographic location	Northern England (Midsize industrial town)		
Language	English		
Method of data collection	Interviews, Group discussions		
Method of analysis	Thematic analysis		
Setting	3 hospitals (old hospital and ward of a general hospital moved to a new hospital)		
Study Period	2010–2011		
Sample	Carers (*n* = 9)		
Recruitment method	Purposive sampling		
Purpose of the study	“(…) use the analytical device of the carer’s ‘journey’ to explore the extent to which carers seem to be positioned as ‘outsiders’ in the hospital space, the degree to which they experience the hospital space as ‘permeable’ and their individually variable and contingent sense of whether the hospital provides a ‘therapeutic landscape’”. (Wood et al. [51] (p. 123))		

Notes. NA: not available.

**Table 4 ijerph-19-01490-t004:** Reasons for exclusion.

Reason	Number of Studies
Wrong setting or concerns regarding the setting	76
Wrong sample	5
No mental health outcome investigated	18
No environmental stimuli	25
Specific therapeutic approach	7
No relationship investigated between features of a TL and MH outcomes	32
Methodological concerns	23
Wrong type of paper	26
Study did not meet any inclusion criteria (e.g., validation of questionnaires)	23
Study not available	2

**Table 5 ijerph-19-01490-t005:** Methodological quality of the included studies.

	Agrest et al. [39]	Donald et al. [40]	Gilburt et al. [41]	Hung et al. [42]	McGonagle and Allan [43] ^a^	Muir-Cochrane et al. [44]	Nanda et al. [45] ^b^	Novotná et al. [46]	Schröder and Ahlström [47]	Shepley et al. [48]	Simonsen and Duff [49]	Wood et al. [51]	Wood et al. [50]
Research design overview	▲	▲	▲	▬	-/-	▼	▬	▬	▬	▬	▬	▬	▬
Description of the research design	▲	▬	▲	▬	-/-	▬	▬	▬	▬	▬	▼	▬	▬
Rational for the selected design	▲	▲	▲	▲	-/-	▼	▲	▲	▲	▲	▲	▲	▲
Study participants or data sources	▲	▬	▲	▬	-/-	▬	▼	▬	▲	▬	▬	▬	▬
Numbers of participants/documents/events	▲	▲	▲	▲	-/-	▲	▬	▬	▲	▲	▬	▬	▬
Description of participants/data sources	▲	▼	▲	▬	-/-	▼	▼	▬	▲	▬	▬	▬	▬
Researcher characteristics	▲	▼	▲	▼	-/-	▼	▼	▼	▼	▼	▼	▼	▼
Researcher description	▲	▼	▲	▬	-/-	▬	▼	▼	▼	▼	▼	▬	▬
Researcher-participant relationship	▲	▼	▲	▼	-/-	▼	▼	▬	▼	▼	▼	▼	▼
Participant recruitment	▬	▼	▬	▬	-/-	▬	▼	▼	▼	▼	▼	▬	▼
Recruitment process	▬	▼	▬	▬	-/-	▬	▬	▼	▬	▬	▬	▬	▼
Participant selection	▬	▼	▬	▬	-/-	▬	▼	▬	▼	▼	▼	▬	▬
Data collection	▬	▬	▬	▬	-/-	▬	▬	▬	▬	▬	▬	▬	▬
Data collection and identification procedures	▬	▬	▬	▬	-/-	▬	▬	▬	▬	▬	▬	▼	▬
Recording and data transformation	▲	▲	▲	▲	-/-	▲	▲	▲	▲	▬	▲	▲	▲
Analysis	▬	▬	▬	▬	-/-	▬	▼	▬	▬	▬	▼	▬	▬
Rigour and transparency of data-analytical strategies	▬	▬	▬	▬	-/-	▬	▼	▬	▬	▬	▼	▬	▬
Methodological integrity	▲	▬	▲	▲	-/-	▬	▬	▬	▲	▬	▬	▲	▲
Overall	▬	▬	▬	▬	-/-	▬	▼	▬	▬	▬	▼	▬	▬

Notes. ▲ high quality; ▬ medium quality; ▼ low quality ^a^ quantitative study, methodological quality not assessed ^b^ assessment of the qualitative study part only.

## Data Availability

Not applicable.

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
