# Peer review of "Therapeutic Landscapes and Psychiatric Care Facilities: A Qualitative Meta-Analysis"

_ijerph, 2022, doi:10.3390/ijerph19031490_

Round 1

Reviewer 1 Report

 The manuscript 'Therapeutic landscapes and psychiatric care facilities: A qualitative meta-analysis' is an important and solid qualitative study that provides relevant information on environmental healthcare facilities promoting health and recovery in psychiatric services for various stakeholders and policymakers. The study is well documented by the articles cited and its methodology is rigorous.

However, I have a few suggestions to improve the manuscript: 

  1. Introduction - the research topic should be integrated into the broad context of the UN Convention on the Rights of Persons with Disabilities (CRPD) (United Nations, 2006). 

2. Discussion - findings should be addressed with evidence-based criteria for a beneficial therapeutic landscape. Examples are relevant factors proposed by the WHO QualityRights initiative (Pathare et al., 2021), or by the European Psychiatric Association guidance on quality assurance in mental healthcare (Gaebel et al., 2015).

Specific comments:

  1. Section 1.3. Please specify the criteria for each of the six categories of Curtis et al. (2007). (Lines 72-78).
  2. Full names/expressions should be provided with the first use of abbreviations, for example, PICO (Line 93).
  3. Lines 111-112 and 116-126 – Replace 'two authors' and [Anonymous 1] by the initials of the names of the relevant authors.
  4. Section 3.2. Please specify how 'medium quality' was determined (Lines 237-238).
  5. Conclusion –The practical recommendations for decision-makers should be extended.

Reviewer 2 Report

This systematic review "Therapeutic landscapes and psychiatric care facilities: A qualitative meta-analysis" reviewed the effects of therapeutic landscapes for different stakeholders in psychiatric care facilities". The authors evaluated 13 articles out of 2853.

I would like to congratulate the authors, who, despite having only 13 articles, managed to summarize important information about mental health and service users, TL, as well as stakeholders.
Light and well-organized writing.

Five suggestions:
a) review the keywords that are already in the title of the article;
b) Lines 106-109, the reason for the exclusion is not very clear. Please make it clearer;
c) Table 1, were reviews excluded? If yes, insert in the Exclusion criteria;
d) Figure 1: about "Additional records identified through other sources", which other source?;
e) Explore discussion about TL. I thought they could get rich.
